# Psychometric Properties of the Maternal and Paternal Parenting Styles Scale in Chilean Adolescents

**DOI:** 10.3390/ijerph18126229

**Published:** 2021-06-09

**Authors:** José Luis Gálvez-Nieto, Karina Polanco-Levicán, Braulio Navarro

**Affiliations:** 1Departamento de Trabajo Social, Universidad de La Frontera, Temuco 4780000, Chile; jose.galvez@ufrontera.cl; 2Departamento de Psicología, Universidad Católica de Temuco, Temuco 4780000, Chile; 3Carrera de Pedagogía en Educación Física, Universidad Autónoma de Chile (Chile), Temuco 4780000, Chile; 002ademir@gmail.com

**Keywords:** parenting styles, parental socialization, psychometric properties

## Abstract

Parenting style has been related to a series of positive outcomes that extend into adulthood. The aim of this study was to analyze the psychometric properties of the maternal and paternal parenting styles scale (PSS-MP) in a sample of adolescents. A cross-sectional design was used, with a probability sample of 2683 adolescents (48.8% women) from 32 public, private, and subsidized schools in Chile. In total, four confirmatory factor models were contrasted, which was the best fit to support the originally proposed six-correlated factor structure. The factor invariance analysis reflected that the metric equivalence according to sex is present at the level of scale invariance. It is concluded that the abbreviated version of the PSS-MP provides sufficient evidence for use in the Chilean adolescent population.

## 1. Introduction

Parenting styles are a highly relevant construct for both academia and society in general. Different studies have demonstrated the impact that parenting style has from the earliest stages to adulthood [1,2]. A definition of parenting style conceptualizes it as a set of attitudes and childcare patterns that generate an emotional climate in the parent/child relationship [3].

The field of parenting style research distinguishes two theoretical approaches. From the classic model comes the typological approach, based on the crossing of the variables of affect and control [4]. From these dimensions four typologies of parenting style emerge: democratic, characterized by high levels of control and affect; negligent, low levels of control and affect; permissive, low level of control and high level of affect; and authoritarian, defined by high levels of control and low levels of affect. Using the multidimensional models, this approach considers different latent variables shaping parenting style and puts them in relation to other facets of the adjustment or competence of their children [5].

With respect to the multidimensional approach, in addition to the variables of affect and control, other ways to measure the relationship between parents and children appear, such as the parenting styles assessment scale [6]. This instrument proposes a six-dimension structure: affect and communication, behavioral and psychological control, promotion of autonomy, revelation, and humor. Subsequently, [7] developed a typology of parenting styles, resulting in three styles, called democratic, strict, and indifferent. The democratic style is characterized by affect, the promotion of autonomy, revelation, good humor, and little psychological control. Behavioral control was not considered a relevant dimension; on the contrary, the revelation of information allows fathers and mothers to achieve greater control over their child’s behavior. The strict style shows lower levels of affect than the democratic style, which occurs with revelation, humor, and promotion of autonomy; however, there is a high level of psychological and behavioral control. The indifferent style was characterized by low scores in all the previously mentioned dimensions, except in psychological control [8].

The affect/communication dimension turned out to be the most relevant because it is associated with indicator variables of good adolescent adjustment [6]. Of the studies conducted using this scale, it is noted that psychological control is associated with greater psychopathological intensity and an externalizing group of symptoms [9]. By contrast, the greater the affect, behavioral control, revelation, and humor, the fewer the psychopathological symptoms [10].

Other authors [11] indicated that with greater affect, communication, autonomy, behavioral control, humor, and revelation, the offline school aggression, antisocial behavior, and friendships that present antisocial behaviors decrease. On the other hand, the lower behavioral control associated with the permissive style is linked to teen pregnancy [12]. Meanwhile, adequate behavioral control and revelation are positively and significantly related to the time dedicated to study and academic performance [13].

It is relevant to consider that this scale differentiates between maternal and paternal parenting styles, which enriches, expands, and leads to the information given by the adolescent about their parents being more pertinent. As [6] illustrated, there are differences between children’s perception of their mothers and fathers; they referred to teenagers perceiving their mothers as affectionate but at the same time more controlling. On the other hand, it is suggested that among the protective factors against aggressiveness, behavioral control in the father and revelation in the mother are considered relevant [14]. Other authors [15] also suggested that the mother’s affection and communication and the father’s promotion of autonomy are protective factors against bullying.

The current study is based on a multidimensional measurement of the PSS-MP Scale assessing paternal and maternal parenting styles. This self-application scale was originally designed in Spain, and aims to assess parenting styles based on the teenager positive development approach [16]. According to its factorial structure, this instrument assesses six dimensions: affect and communication, behavioral and psychological control, promotion of autonomy, revelation, and humor.

Regarding applications of the PSS-MP scale, it has shown acceptable psychometric properties in diverse samples; e.g., a recent study confirmed its factorial structure in 1507 Spanish students [17]. In another piece of research, a different study analyzed its psychometric properties in an abridged version of 24 questions, reducing the number of items and maintaining the six correlated factors [11].

Besides psychometric studies, the PSS-MP scale has shown statistically significant relations with life satisfaction [18,19], extracurricular activities participation, and positive personal development [20]. The psychological control dimension associated to a higher degree with psychopathology and externalizing symptomatology [9]. On the contrary, the higher the affect, behavioral control, and humor, the lower the psychopathological symptoms [10].

Considering the relevance of evaluating parenting styles and the possibility of measuring this construct from a multidimensional perspective, the first hypothesis of this study posits that the scores on the PSS-MP will maintain a six-correlated factor structure in addition to suitable levels of reliability. The second hypothesis is that the scale will continue to be equivalent until the level of scalar invariance for the variable sex. Therefore, the aim of this study was to analyze the psychometric properties of the maternal and paternal parenting styles scale (PSS-MP) in a sample of adolescents. The second objective was to analyze the degree of invariance according to sex.

## 2. Materials and Methods

### 2.1. Participants

The population was comprised of (N) 486,427 adolescent students from public, private, and subsidized secondary schools belonging to five representative regions in the Macrozones of Chile. The participants were selected through stratified, multi-stage probability sampling, with a reliability of 99.7%, a 3% margin of error, and a variance p = q = 0.5 [21]. Three variables were considered for the stratification: region, type of school, and type of teaching at the school. The sample was comprised of (n) 2683 students from 32 schools, of both sexes (48.8% women), and with an average age of 15.78 (SD = 1.35). The data were collected in the second half of 2019. The selected schools included students of various socioeconomic levels, but mainly represented low and middle socioeconomic levels.

### 2.2. Instruments

A sociodemographic questionnaire was applied to characterize the students. The Appendix A contained the following closed questions: sex, age, region, family origin (urban/rural), type of teaching, type of school, and grade.

In addition, the maternal and paternal parenting styles scale (PSS-MP) was applied. This scale is a self-report instrument [6] that assesses parenting styles from the teenager’s perspective using 82 items on a 6-point Likert-type scale (41 items for the father and 41 items for the mother). The PSS-MP has six domains: affect and communication (8 items, e.g., “He/She enjoys talking to me”), promotion of autonomy (8 items, e.g., “He/She encourages me to express my ideas although other people do not like them”), behavioral control (6 items, e.g., “He/She wants to know where I am going when I go out”), psychological control (8 items, e.g., “He/she is not so nice to me when I do not do things his/her way”), revelation (5 items, e.g., “I talk to him/her about problems I have with my friends”), and humor (6 items, e.g., “He/She is almost always happy and optimistic”). The scale presented adequate psychometric indicators in the original validation study [6].

### 2.3. Procedure

Contact was made with the school principals, and authorization was requested to apply the questionnaires. With the aim of protecting the ethical principles of the project, informed consent was sought from mothers, fathers, guardians, and students. The study was approved by the Ethics Committee of the Universidad de La Frontera (Ethics protocol number 034-19). The questionnaires were answered anonymously during the first period of class.

### 2.4. Data Analysis

Missing values were less than 5% of the sample and treated with the multiple imputation method available in the MPLUS v.8.1 software [22]. In addition, descriptive statistics, measures of central tendency, dispersion, and shape were estimated using SPSS version 23.0. Then, the structure of the scale was evaluated using confirmatory factor analysis (CFA) models. For the implementation of the CFA models, the polychoric correlations matrix and the weighted least squares means and variance adjusted (WLSMV) estimation methods were used. For the goodness-of-fit indices (comparative fit index (CFI), Tucker–Lewis index (TLI)) values greater than or equal to 0.90 were considered reasonable [23], and for root mean square error of approximation (RMSEA) values less than or equal to 0.08 were considered reasonable [24]. Then, a factor invariance analysis was performed for the variables of sex, type of education, and age. This analysis considers the following models [25]: M0 configural (equal number of factors), M1 metric (equal factor loadings), and M2 scalar (equality of intercepts). Reliability was evaluated with the following estimators: McDonald’s Omega and Cronbach’s alpha [26].

## 3. Results

### 3.1. Descriptive Analysis and Confirmatory Factor Analysis

The results presented below are based on the analysis of 2683 adolescent students. Table 1 presents the descriptive statistics of the PSS-MP. The highest mean for the fathers’ scores was obtained by item 19, “When I go out on Saturday night I have to tell him beforehand where I am going and when I will be back” (M = 5.24, DT = 1.45) and the highest mean for the mothers was obtained by item 7, “If I have a problem, I can count on her help” (M = 5.44, DT = 1.05).

With respect to the factorial structure of the PSS-MP, two CFA models were estimated with the 82 items on the scale (41 items for the father and 41 items for the mother). The first estimated model was for the scores obtained by the mothers (WLSMV-χ^2^ (df = 764) = 5842.788; CFI = 0.942; TLI = 0.938; RMSEA = 0.057 (C.I = 0.056–0.059)) and the second model was for the fathers (WLSMV-χ^2^ (df = 764) = 7264.536; CFI = 0.935; TLI = 0.930; RMSEA = 0.072 (C.I = 0.071–0.074)). Both models presented acceptable goodness-of-fit indices. The factor loadings presented satisfactory and statistically significant results for both the father and the mother (Table 1). These results contribute evidence supporting the six-correlated factor model as presenting a good fit to the data.

### 3.2. Factorial Invariance

As Table 2 shows, the degree of factorial invariance was evaluated between men and women, for mothers and fathers. The first model tested was the M0 or configuration invariance, in both cases the parameters were statistically significant, confirming that the six-factor structure is stable according to sex.

Then, the M1 model was evaluated, which added restrictions to the factor loadings. The results indicate that there are statistically significant differences between the metric and configuration models, between men and women. For the scale for mothers (*p* [ΔWLSMV−χ^2^] < 0.001), these results make it possible to reject the metric equivalence between men and women; therefore, the following nesting level for the mothers will not be continued. With respect to the results for the scale on fathers, no statistically significant differences were found between the metric and configuration models between men and women (*p* [ΔWLSMV−χ^2^] = 0.0267). Based on these results, the M2 model, called scalar invariance, was evaluated, which added constraints to the intercepts. This model showed statistically significant differences between the scalar and metric models, between men and women, for the scale on the father (*p* [ΔWLSMV−χ^2^] < 0.001); therefore, equivalence between the two models was rejected.

### 3.3. Evidence of Reliability

Table 3 provides evidence for the reliability of the scale, considering the six-correlated factor model. The reliability indices provided an adequate reliability for each factor, for both the mother and the father, noting that the factor affect and communication presented the highest values for both the mothers (ω = 0.931; α = 0.921) and the fathers (ω = 0.948; α = 0.948).

## 4. Discussion

This investigation had two aims. The first was to analyze the psychometric properties of the maternal and paternal parenting styles scale (PSS-MP) in a sample of Chilean adolescents. The second was to analyze the degree of metric invariance according to sex. The results of this study support the first hypothesis, which posited that PSS-MP scores maintain a six-correlated factor structure in addition to suitable levels of reliability. These findings confirm the presence of a theoretical structure consistent with the original study [6]. 

Regarding the fulfillment of the second hypothesis, the results demonstrated that the scores on the adapted version of the PSS-MP remain steady until the level of metric invariance for the sex variable. It also provides a measurement of parenting style with robust theoretical support. As for the content of the instrument, the scale presents consistent coverage based on a six-correlated factor structure, and relying on the positive adolescent development approach. The findings of this study show that the PSS-MP scale is a tool that will contribute valid and reliable information for decision-making in the areas of theoretical research and interventions in the area of application.

Future lines of enquiry could assess relations to other constructs related to the field of primary health care, such as family functioning [27] or in the context of mental health in schools, for example [28,29]. It is suggested that psychometric studies be continued, in order to contribute evidence of convergent validity and to analyze the invariability of the construct in other student populations, so as to facilitate the comparison of the results between different sociocultural contexts.

## 5. Conclusions

Given the importance of the family as a social institution, measuring parenting styles is key to understanding adolescent development because they are a fundamental element in evaluating the quality of family relationships. This research aimed to analyze the psychometric properties of the maternal and paternal parenting styles scale (PSS-MP) in a sample of Chilean adolescents. The results showed an adequate adjustment in terms of re-liability and validity. In addition, a factorial invariance analysis was performed to evaluate the quality of the measurement according to sex. The relevance of these findings is the following: to provide a reliable and valid measurement of parenting style to be used in the Chilean context, and to provide a tool linked to the promotion of health, substance use, and risk behaviors, which can be influenced by maternal and paternal parenting styles.

## Figures and Tables

**Table 1 ijerph-18-06229-t001:** Descriptive statistics and confirmatory factor analysis.

	Father					Mother				
Items	Mean	Standard Deviation	Asymmetry	Kurtosis	CFA ^1^	Mean	Standard Deviation	Asymmetry	Kurtosis	CFA
It1	4.80	1.39	−1.29	0.98	0.864 *	5.28	1.03	−1.81	3.68	0.862 *
It2	4.48	1.47	−0.87	−0.10	0.829 *	5.09	1.15	−1.41	1.58	0.850 *
It3	4.83	1.41	−1.30	0.94	0.799 *	5.22	1.09	−1.70	2.98	0.855 *
It4	4.80	1.52	−1.25	0.55	0.874 *	5.21	1.22	−1.74	2.53	0.877 *
It5	4.33	1.65	−0.77	−0.56	0.804 *	4.93	1.37	−1.36	1.14	0.833 *
It6	4.55	1.60	−0.99	−0.12	0.857 *	4.96	1.33	−1.40	1.32	0.882 *
It7	5.07	1.43	−1.64	1.80	0.856 *	5.44	1.05	−2.30	5.40	0.864 *
It8	4.74	1.51	−1.22	0.51	0.855*	5.15	1.24	−1.72	2.53	0.884 *
It9	4.33	1.62	−0.77	−0.53	0.825 *	4.73	1.43	−1.12	0.43	0.863 *
It10	4.64	1.55	−1.08	0.14	0.784 *	4.94	1.33	−1.37	1.20	0.824 *
It11	4.71	1.55	−1.15	0.27	0.773 *	5.04	1.31	−1.56	1.88	0.840 *
It12	4.82	1.48	−1.32	0.86	0.777 *	5.04	1.29	−1.56	1.94	0.827 *
It13	4.93	1.43	−1.42	1.16	0.734 *	5.13	1.21	−1.63	2.37	0.801 *
It14	4.60	1.60	−1.06	0.01	0.707 *	4.89	1.37	−1.32	1.03	0.778 *
It15	4.51	1.58	−0.93	−0.21	0.728 *	4.76	1.43	−1.17	0.56	0.790 *
It16	4.69	1.56	−1.14	0.23	0.759 *	4.95	1.39	−1.45	1.33	0.796 *
It17	5.22	1.36	−1.97	3.03	0.857 *	5.66	0.84	−3.38	12.96	0.906 *
It18	5.11	1.47	−1.76	2.02	0.787 *	5.58	0.94	−2.98	9.71	0.882 *
It19	5.24	1.45	−2.01	2.83	0.818 *	5.61	0.98	−3.28	11.15	0.906 *
It20	4.42	1.60	−0.86	−0.37	0.858*	4.86	1.39	−1.31	1.03	0.863 *
It21	4.70	1.68	−1.16	0.04	0.585*	5.10	1.38	−1.64	1.82	0.739 *
It22	3.99	1.82	−0.47	−1.17	0.541 *	4.43	1.68	−0.87	−0.47	0.680 *
It23	3.86	1.75	−0.37	−1.16	0.530 *	4.04	1.70	−0.51	−0.98	0.600 *
It24	3.95	1.66	−0.38	−1.00	0.553 *	4.29	1.57	−0.60	−0.69	0.644 *
It25	3.11	1.87	0.26	−1.40	0.739 *	3.26	1.89	0.14	−1.46	0.760 *
It26	3.01	1.75	0.31	−1.24	0.804 *	3.08	1.80	0.26	−1.32	0.792 *
It27	3.85	1.81	−0.29	−1.30	0.709 *	4.02	1.76	−0.42	−1.15	0.764 *
It28	2.16	1.59	1.16	0.06	0.818 *	2.25	1.63	1.06	−0.20	0.753 *
It29	2.58	1.80	0.73	−0.93	0.818 *	2.71	1.85	0.60	−1.15	0.783 *
It30	3.13	1.85	0.25	−1.37	0.650 *	3.31	1.85	0.10	−1.42	0.675 *
It31	3.70	1.81	−0.24	−1.31	0.842 *	4.12	1.75	−0.57	−0.97	0.837 *
It32	3.80	1.81	−0.34	−1.25	0.800 *	4.29	1.69	−.073	−0.69	0.858 *
It33	3.16	1.87	0.21	−1.42	0.786 *	3.91	1.89	−0.35	−1.35	0.777 *
It34	3.78	1.83	−0.29	−1.32	0.839 *	4.41	1.68	−0.81	−0.58	0.846 *
It35	3.56	1.89	−0.12	−1.44	0.719 *	4.04	1.83	−0.50	−1.15	0.759 *
It36	4.67	1.40	−1.14	0.65	0.811 *	4.83	1.28	−1.23	1.11	0.836 *
It37	4.72	1.41	−1.11	0.53	0.834 *	4.91	1.24	−1.24	1.17	0.889 *
It38	4.99	1.43	−1.48	1.31	0.822 *	4.93	1.36	−1.32	1.03	0.847 *
It39	4.89	1.45	−1.37	1.00	0.909 *	5.05	1.27	−1.49	1.74	0.919 *
It40	4.76	1.51	−1.18	0.40	0.898 *	5.03	1.29	−1.42	1.44	0.917 *
It41	4.47	1.51	−0.88	−0.13	0.744 *	4.48	1.45	−0.860	−0.05	0.736 *

^1^ Confirmatory factor analysis. * = *p* < 0.01.

**Table 2 ijerph-18-06229-t002:** Factorial invariance according to sex.

		WLSMV−χ^2^ (df)	CFI	TLI	RMSEA	Comp.	ΔWLSMV−χ^2^	Δgl	*p*-Value (ΔWLSMV−χ^2^)	ΔCFI
Mother	1 M0	6449.354 (1528)	0.946	0.942	0.056					
	2 M1	6445.498 (1563)	0.947	0.944	0.056	2 vs. 1	88.774	35	<0.001	0.001
	3 M2	6684.459 (1721)	0.946	0.948	0.053	3 vs. 2	516.410	158	<0.001	−0.001
		**WLSMV−χ^2^ (df)**	**CFI**	**TLI**	**RMSEA**	**Comp.**	**ΔWLSMV−χ^2^**	**Δgl**	***p*-Value (ΔWLSMV−χ^2^)**	**ΔCFI**
Father	1 M0	7740.223 (1528)	0.939	0.934	0.071					
	2 M1	7776.038 (1563)	0.939	0.936	0.070	2 vs. 1	52.891	35	0.0267	0
	3 M2	8088.790 (1721)	0.937	0.940	0.067	3 vs. 2	431.987	158	<0.001	−0.002

**Table 3 ijerph-18-06229-t003:** Evidence of reliability.

	Father		Mother	
Factors	McDonald’s ω	Cronbach’s α	McDonald’s ω	Cronbach’s α
Affect and communication	0.948	0.948	0.931	0.931
Promotion of autonomy	0.925	0.924	0.892	0.892
Behavioral control	0.899	0.889	0.814	0.811
Psychological control	0.861	0.860	0.854	0.852
Revelation	0.875	0.874	0.857	0.857
Humor	0.924	0.923	0.907	0.907

## Data Availability

No appliable.

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
