# Peer review of "Psychometric Properties of the Maternal and Paternal Parenting Styles Scale in Chilean Adolescents"

_ijerph, 2021, doi:10.3390/ijerph18126229_

Round 1
Reviewer 1 Report
It has been a pleasure to review this manuscript.
The topic is very interesting. I agree with the authors that parental educational styles are a fundamental element in evaluating the quality of family relationships. Being able to measure this construct is the key to understanding adolescent development.
However, I am going to make some comments with the sole objective of improving the quality of the manuscript:
- The introduction would need to better explain the PSS-MP scale. It is not said who developed it, or when or where.
- The adaptation of this scale has already been previously validated in other countries. It would be advisable to explain a little how it has worked in other places.
- In the methodology section, it would be necessary to explain when the data was collected and how long it took to collect it.
- In general, the methodology seems correct to me. The results are adequate and the discussion and conclusion also seem correct to me.
Thanks
Kind Regards
Author Response
We appreciate the comments of the revisor since they have improved the quality of our writing.
1.- We have included a better explanation of the PSS-MP Scale in the introduction.
2.- We have improved the introduction by including previous validations in other countries. Furthermore, other empirical studies have been included for they show further applications of the instrument.
3.- Dates and application time have been included in the methodology section under the instruments’ subsection.
4.- We truly appreciate the comments made by the revisor.

Reviewer 2 Report
This brief report is generally well-written although some English editing is required.
My concerns mostly concern the introduction section. To my opinion, it is not clear enough at this stage. Several concepts have been introduced but need to be treated more in depth, and be contextualized. Please, read my comments below:
- Did you assume that parenting styles and parental education styles are the same? If yes, please maintain consistency in the use of terms or, otherwise, this needs to be specified.
- Please, p. 1 line 38 add: ‘affect and communication’.
- It is not clear to me if indifferent style is characterized by lower (or generic low) scores in all dimensions than the other two styles. I think this needs clarification, maybe a table or figure could help.
- Please, when you describe links between parenting dimensions and adolescent adjustment, it seems a bit confusing. Did you consider the classic model, the multidimensional one, or single dimensions? And if single dimensions were considered, you need to contextualize them making it clear to which model you are referring to.
- P. 2 line 16, I think the term ‘complicates’ is inappropriate. Please substitute it.
Method: p. 2 lines 80-88: the sociodemographic characteristics of the sample should be integrated within results section.
Procedure: p. 3 lines 105-109: please, can you provide the Ethics protocol reference number?
Author Response
Point 1: This brief report is generally well-written although some English editing is required.

Response 1: We appreciate the comments of the evaluator since they have significantly improved the quality of our manuscript. We have revised and improved the English edition; changes have been highlighted.
Point 2: Did you assume that parenting styles and parental education styles are the same? If yes, please maintain consistency in the use of terms or, otherwise, this needs to be specified.
Response 2: Parenting styles have been standardised.
Point 3: Please, p. 1 line 38 add: ‘affect and communication’.
Response 3: The suggestion has been included.
Point 4: It is not clear to me if indifferent style is characterized by lower (or generic low) scores in all dimensions than the other two styles. I think this needs clarification, maybe a table or figure could help.
Response 4: We thank the comment of the revisor. In preliminary reports, he “indifferent style” typology is characterised by the lowest scores. This has been clarified in the body of the manuscript, including the citation that supports this statement.
Point 5: Please, when you describe links between parenting dimensions and adolescent adjustment, it seems a bit confusing. Did you consider the classic model, the multidimensional one, or single dimensions? And if single dimensions were considered, you need to contextualize them making it clear to which model you are referring to.
Response 5: We have clarified that this article considers the multidimensional models.
Point 6: P. 2 line 16, I think the term ‘complicates’ is inappropriate. Please substitute it.
Response 6: Thank you very much, we have changed the concept.
Point 7: Method: p. 2 lines 80-88: the sociodemographic characteristics of the sample should be integrated within results section.
Response 7: Thank you very much. Socio-demographic characteristics were included in the results section.
Point 8: Procedure: p. 3 lines 105-109: please, can you provide the Ethics protocol reference number?
Response 8: We have added the reference number of the ethics protocol.
